# Reanalysis of Gene Expression Profiles of CD4+ T Cells Treated with HIV-1 Latency Reversal Agents

**DOI:** 10.3390/microorganisms8101505

**Published:** 2020-09-30

**Authors:** Antonio Victor Campos Coelho, Ronald Rodrigues de Moura, Sergio Crovella

**Affiliations:** 1Federal University of Pernambuco, Avenida da Engenharia, Cidade Universitária, Recife 50670-901, Brazil; 2Institute for Maternal and Child Health—IRCCS Burlo Garofolo, 34137 Trieste, Italy; ronaldmoura1989@gmail.com (R.R.d.M.); crovelser@gmail.com (S.C.); 3Department of Medical, Surgical and Health Sciences, University of Trieste, 34127 Trieste, Italy

**Keywords:** reservoir, shock-and-kill, antiretroviral therapy, RNA-Seq, transcriptomics

## Abstract

The human immunodeficiency virus (HIV-1) causes a progressive depletion of CD4+ T cells, hampering immune function. Current experimental strategies to fight the virus focus on the reactivation of latent HIV-1 in the viral reservoir to make the virus detectable by the immune system, by searching for latency reversal agents (LRAs). We hypothesize that if common molecular pathways elicited by the presence of LRAs are known, perhaps new, more efficient, “shock-and-kill” strategies can be found. Thus, the objective of the present study is to re-evaluate RNA-Seq assays to find differentially expressed genes (DEGs) during latency reversal via transcriptome analysis. We selected six studies (45 samples altogether: 16 negative controls and 29 LRA-treated CD4+ T cells) and 11 LRA strategies through a systematic search in Gene Expression Omnibus (GEO) and PubMed databases. The raw reads were trimmed, counted, and normalized. Next, we detected consistent DEGs in these independent experiments. AZD5582, romidepsin, and suberanilohydroxamic acid (SAHA) were the LRAs that modulated most genes. We detected 948 DEGs shared by those three LRAs. Gene ontology analysis and cross-referencing with other sources of the literature showed enrichment of cell activation, differentiation and signaling, especially mitogen-activated protein kinase (*MAPK*) and Rho-GTPases pathways.

## 1. Introduction

The human immunodeficiency virus (HIV-1) is the causative agent of acquired immunodeficiency syndrome (AIDS), characterized by a chronic, progressive depletion of CD4+ T cells, hampering immune function and thus posing vulnerability to opportunistic infections [1,2,3]. Antiretroviral therapy (ART) is the standard treatment against human immunodeficiency virus type 1 (HIV-1). Since 1996, it has been saving millions of lives by inhibiting viral genome replication and protein maturation in those infected [4].

Unfortunately, ART does not cure definitively HIV-1 infection because this retrovirus integrates its genome on the cells it infects, setting up latent infection. Should ART treatment be discontinued, the virus is promptly reactivated, restarting production of viral particles and immune system destruction [5]. The major component of the latent viral reservoir are memory CD4+ T cells, which are capable of self-renewal and long-term survival [6]. It has been estimated that memory CD4+ T cells have a mean half-life of around 44 months. Because of this, it would take more than 70 years for the latent reservoir to disappear in an average person in viral suppression [7].

According to two impressive reports of a HIV-1 clinical cure of two men following allogeneic haemopoietic stem-cell transplantation [8,9], the pathway to a definite HIV-1 cure has been paved: the elimination of the latent viral reservoir. The current challenge is therefore the development of a less invasive method than bone marrow transplantation for the elimination of the said reservoir.

A strategy is the search for latency reversal agents (LRAs), which are molecules or compounds that are able to activate latent HIV-1 from its dormant state in the viral reservoir from an infected individual, with the purpose of gradually eliminating the viral reservoir, thus making the virus detectable by the immune system, a method nicknamed as “shock-and-kill” or “kick-and-kill”. The investigated LRAs have generally targeted epigenetic pathways (such as histone deacetylase inhibitors – HDACi, such as vorinostat, also known as suberanilohydroxamic acid or SAHA, and valproic acid) and T cell receptor (TCR) agonists. A major problem of this effort is that LRAs generally fail to reverse latency in studies of cells from HIV-1-infected individuals [10].

We hypothesize that if the molecular pathways elicited by the presence of LRAs are known, perhaps it may help the design of new, more efficient, and safer, “shock-and-kill” strategies. 

Transcriptional profiling via RNA sequencing (RNA-Seq) has been widely used over the last decade and has become the assay of choice to unravel the relationship between cellular function and phenotypes [11,12,13]. Thus, the objective of the present study is reevaluating RNA sequencing (RNA-Seq) assays deposited in public databases to find differentially expressed genes (DEGs) during latency reversal, aimed at unifying knowledge about pathways possibly involved in the action of LRAs on reservoirs from HIV-infected individuals.

## 2. Materials and Methods 

### 2.1. Study Search Strategies

We searched for latency reversal RNA-Seq studies via Sequence Read Archive (SRA) [14] searching through Gene Expression Omnibus (GEO) database [15,16] and PubMed [17] databases. In SRA database, the following search string containing commonly used LRAs [18] was used: HIV-1 AND (latency OR latency reversal agent* OR latency reversing OR LRA* OR romidepsin OR panobinostat OR Q1 OR ingenol-3-angelate OR bryostatin-1 OR PMA OR ionomycin) AND homo sapiens [Organism]. This resulted in 676 abstracts. After filtering for “Expression profiling by high throughput sequencing” in Study type, we obtained 18 abstracts.

A similar search string: HIV-1 AND (OR romidepsin OR panobinostat OR Q1 OR ingenol-3-angelate OR bryostatin-1 OR PMA OR ionomycin) AND homo sapiens AND RNA-Seq, was used in PubMed database, resulting in 15 abstracts. The list of abstracts retrieved by the two strategies was downloaded and processed in a spreadsheet to remove duplicates and for further evaluation if they would be included in the study.

For inclusion in the reanalysis, we used the following criteria: whole transcriptome studies; experiments carried out in CD4+ T cells and availability of the raw data (fastq files) for each sample. The search was performed on 16 March 2020.

### 2.2. RNA-Seq Data Collection, Processing, and Statistical Analyses

Raw fastq files were downloaded using *SRAdb* package [19] for R software v. 3.6.1 [20] or via Phil Ewel’s SRA-Explorer online tool (http://sra-explorer.info/). Only sequencing reads from samples meeting the following criteria were downloaded: LRA-treated and negative controls (not treated with LRA or vehicle-treated, such as DMSO-treated cells). Exclusion criteria: mock-infected samples, single cell samples, positive controls samples (such as CD3+CD28+IL-2 stimulation to induce global CD4+ T cell activation, which is toxic and not clinically relevant). To simplify analyses, we downloaded the runs for the last time point only in case the original study had time-repeated design.

The reads were re-processed using Trimmomatic software v. 0.39 [21] to trim Illumina adapters and to exclude reads counting less than 25 bases. Then, the remaining reads were mapped on the National Center for Biotechnology (NCBI) human GRCh38 reference genome and sorted by coordinates using STAR aligner [22]. Binary Alignment Map reads (BAM files) were imported into R software and processed with Rsubread package [23], whose featureCounts function mapped sequencing reads to genomic features using an in-built human GRCh38 genome annotation (28.395 genes), quantifying raw expression levels per gene per sample, producing a gene count table for each sample. All subsequent analyses were made with R software v. 3.6.1 [20].

The counts from technical replicas and CD4+ T cell subpopulations were collapsed into single count per unique sample, representing gene expression in CD4+ T cell pools. Then, sample counts were normalized via trimmed mean of M values (TMM) method in a study-by-study basis, allowing differential gene expression analysis by negative binomial generalized linear models with per gene, per LRA testing by quasi-likelihood F-test, alongside calculation of log2-transformed expression fold changes for each gene (through comparison of LRA-treated samples with non-LRA-treated samples) with the help of edgeR package [24].

Finally, log2-transformed fold changes mentioned above were matched with their respective pooled, false discovery rate (FDR) adjusted *p*-values. The genes with both |log2(Fold-Change)| > 1 and adjusted *p*-value < 0.05 via were considered to be statistically significant DEGs.

Gene names (symbols) were derived from gene ids with annotate [25] and org.Hs.eg.db [26] packages.

### 2.3. Cross-Referencing and Set Analysis of DEGs

We cross-referenced the differential expression re-analyses results with the observations of other studies from the literature investigating HIV-1 infection and replication [27,28,29,30] and two databases: HIV-1 Human Interaction Database [31,32,33] and RNAcentral, a database of non-coding RNA [34] using NCBI RefSeq [35] gene symbols as external identifiers. Briefly, the tables obtained from the differential expression analyses and curated tables from the two databases were imported into a local PostgreSQL relational database. The tables were linked to reveal which genes were activated by each LRA and more importantly, which genes were modulated by several LRAs, and to assess if said genes were already identified by the literature, so we could identify genes consistently involved in HIV-1 latency reversal.

The VennDiagram [36] package was used for evaluation of intersection among the selected results, revealing the number of commonly modulated genes by different LRAs.

To assess whether our results agreed with original sources investigated, we performed simulations to determine an expected overlap number if each set were truly independent (created by pure chance). The expected overlap number was determined by producing empirical simulations using “mock genes”, a list of random but unique strings to represent gene symbols with the *ids* package of R software [37]. 

First, we produced a list of 28395 unique random strings to represent our annotated human genome. From this list, we randomly sampled two independent sets. The first one contained *R* elements, representing our list of genes following differential expression analysis. The second one contained *O* elements, where *O* was the size of the original genes list reported by each source. The process of formation of these two sets was repeated 10,000 times. Each time, the two sets were different from the previous ones, and the number of overlapping genes (intersection) was calculated.

Thus, we obtained 10000 intersection values. The median of these values was considered to be the expected number. Therefore, the binomial distribution was used to test the observed intersection number with this expected number via one-sided tests under the null hypothesis that the intersection of the original source with the reanalysis list is equal or less than expected by chance. If *p*-value < 0.05, we would reject the null hypothesis and assume that our results had a higher concordance with the source than expected by chance.

### 2.4. Gene Ontology and Reactome Pathway Enrichment Analysis

After obtaining the intersection list as described above, we performed a gene ontology (GO) enrichment analysis through over-representation analyses with the goana function of the limma package [38]. Then, the p-values were FDR-adjusted also with R software. The genes were ranked from the lowest to highest below level of significance α = 0.05.

Additionally, we conducted pathway analysis, based on the REACTOME database, of the statistically significant DEGs using the ReactomePA package [39]. Each pathway *p*-value was FDR-adjusted with *p*-values < 0.05 being considered significant.

## 3. Results

The search strategy resulted in 33 abstracts. Following duplicates removal, abstracts were screened, producing a shortlist of ten studies for further review to check if they met the inclusion criteria mentioned previously. From those, six were reanalyzed, being five studies with published reports [40,41,42,43,44]. We could not ascertain if the remaining project, submitted publicly by the contributor Vallejo-Gracia on April 2019, which was retrieved from the search through GEO database has already been published. We assumed they fit into our inclusion criteria, so we included this project in the reanalysis.

The other four were excluded; three were conducted in uninfected cells [45,46,47] and one did not provide raw sequencing reads in public databases [48].

The first study, from Mohammadi et al. (GSE95297) [40] recruited HIV-1-infected individuals with controlled viremia (plasma viral load, pVL < 20 copies/mL), isolated and pooled their resting CD4+ T cells and treated with disulfiram, interleukin (IL)-7 or SAHA. Since the authors suspended the drugs in dimethyl sulfoxyde (DMSO), the control sample was treated with DMSO only, thus totaling four samples selected: three treated samples and one control sample.

The second study, from Golumbeanu et al. (GSE111727) [41] recruited blood donors and HIV-1 patients included in the Swiss HIV-1 Cohort Study. CD4+ T cells were purified from uninfected blood donors and activated through TCR stimulation. Three days post-activation, cells were transduced with a recombinant HIV-1 vector. Then, cells were allowed to revert to a resting phenotype by long-term culture. Finally, for reactivation, cells were incubated with SAHA for 24 h. Control cells were left untreated. This study contributed with two samples: one untreated and one SAHA-treated.

The third study, from Beliakova-Bethell et al. (GSE114883) [42] used samples of primary CD4+ T cells from HIV-1-negative blood donors which were then infected through a laboratory bystander infection model [49]. This study included 12 samples from four donors, being four controls and eight LRA-treated (one control sample per donor—treated with DMSO only and two treated samples per donor—with SAHA or romidepsin).

The fourth study, from Kulpa et al. (GSE94150) [43] recruited virologically suppressed HIV-1-infected individuals, isolated their CD4+ T cells and subjected them to four conditions: no stimulus (control cells), stimulation for 24 h with: phorbol 12-myristate 13-acetate (*PMA*) plus ionomycin; IL-15 or bryostatin. It is important to highlight that the authors sequenced three CD4+ T cells subpopulations separately (effector memory, transient memory, and central memory) for each donor. For our analysis, however, we pooled together the sequencing reads from those three groups, to approximate a whole CD4+ sample from each donor. This study contributed with nine samples (three controls, one from each donor, three IL-15-treated, and three bryostatin-treated, both treatments also one sample from each donor).

The fifth study, by Vallejo-Gracia et al. (GSE129522) (unpublished), seems to include three donors, which their CD4+ T cells were treated with the LRA AS1842856, a small molecule that acts as a FOXO1 transcription factor inhibitor [50], or received only DMSO as control for at most 48 h, totaling, therefore, six samples.

The sixth study, by Nixon et al. (GSE142774) [44] included three virologically suppressed HIV-1-infected donors. Cells from each patient were treated with DMSO (control), AZD5582, an antagonist of inhibitor of apoptosis proteins (IAPs) [51], or ingenol B, a protein kinase C (PKC) agonist [52] for at most 24 h, resulting in 12 samples (one control and two LRA-treated samples per donor).

In summary, we considered nine strategies in our reanalysis: disulfiram, IL-7, SAHA, romidepsin, IL-15, bryostatin, AS1842856, AZD5582 and ingenol B. Overall, 45 unique samples were selected for inclusion in the reanalysis, being 29 treated with any LRA and 16 were (negative) controls. Table 1 summarizes the studies’ information and main results. Appendix A lists the selected individual samples alongside their respective sequencing reads ids from SRA database.

Overall, there were 15953 statistically significant LRA-differentially expressed gene pairs, corresponding to 9494 unique genes. AS1842856 and IL-15 stimulation did not significantly modulate genes. Appendix A contains this list of LRA-gene pairs, with their respective log2 (Fold-Change) and FDR-adjusted quasi-likelihood F-test *p*-value.

Among the remaining seven LRAs, three remarkedly modulated the most genes: romidepsin (5798 genes, 33.7% of all expressed genes in the samples), followed by AZD5582 (4512, 30.0%) and SAHA, (3218, 19.0%) also in the Beliakova-Bethell et al. [42] study samples. All three were associated with the differential expression of more than 10% of all expressed genes in the sample, whereas all others LRAs modulated much less genes (disulfiram: 473, 2.8%; ingenol B: 443, 3.0%; Il-7: 199, 1.2% and bryostatin: 20, 0.2%; results summarized in Table 2).

There were a few genes shared by LRAs. Most genes (5105 among 9494, 53.8%) were modulated by a single LRA only. Any combination of two, three and four LRAs shared 3089 (32.5%), 1132 (11.9%) and 152 (1.6%) genes, respectively. Just 16 genes (0.2%) where shared by any combination of five LRAs: *ACE, AIRE, BAIAP3, C20orf197, CACNA1I, CR1, CXCR4, DTX4, FCGBP, GPA33, LOC105377746, NME2, PI16, SIRPG-AS1, TCF7* and *TMIE*. No gene was simultaneously activated by six or seven distinct LRAs (Table 3). Appendix A contains the list of 9494 unique genes differentially expressed during LRA stimulation, the DEG list from each study/LRA and the cross-referencing with the literature and databases. Appendix A contains the detailed distribution of how many genes were activated by number of HIV latency reversal agents.

The intersection of the genes modulated by romidepsin, AZD5582 and SAHA were the largest among all latency reactivation strategies, since they caused the differential expression of the most genes. Therefore, we conducted further analyses with the 948 genes modulated in common by those three strategies (“intersection list” thereof; Figure 1).

In general, agreements between the intersection list and other studies directly dealing with HIV-1 infection and replication establishment [27,28,29,30] were small and within the expected by chance, except for the HIV interaction database [31,32,33], whose agreement was higher than expected by chance (*p* < 0.001, Table 4). We observed that 233 genes of the intersection list (24.6% of the intersection list) were already identified in viral-host interaction pathways. Additionally, we detected 110 non-coding RNA genes in the intersection list (11.6%; 108 belonging to lncRNA class and the remaining two belonging to miRNA or antisense RNA classes).

The GO analysis results (with the intersection list as input) showed that 278 pathways from the biological process ontology were enriched after multiple comparison adjustment by FDR. GO terms related to cell activation (GO:0042110 “T cell activation”; GO:0001775 “cell activation”, GO:0002694 “regulation of leukocyte activation”), differentiation (GO:0030154 “cell differentiation”), proliferation (GO:0008283 “cell population proliferation”) and cell signaling (GO:0023052 “signaling”; GO:0007165 “signal transduction”; GO:0023051 “regulation of signaling”) were prevalent among the enriched pathways. Signal transduction pathways were also enriched alongside signaling pathways. with the participation of mitogen-activated protein kinase (*MAPK*) and Rho-GTPases pathways (GO:0043405 “regulation of MAP kinase activity; GO:0043407 “negative regulation of MAP kinase activity”; GO:0007266 “Rho protein signal transduction”; GO:0035023 “regulation of Rho protein signal transduction”). Select enriched GO terms (sorted by FDR-adjusted p-value) are displayed in Table 5; the complete list is displayed on Appendix A. The reactome pathway analysis detected eight enriched pathways in common: R-HSA-8878171 “Transcriptional regulation by *RUNX1*”, R-HSA-3928665 “EPH-ephrin mediated repulsion of cells”, R-HSA-416482 “G alpha (12/13) signaling events”, R-HSA-194840 “Rho GTPase cycle”, R-HSA-8939236 “*RUNX1* regulates transcription of genes involved in differentiation of HSCs”, R-HSA-157118 “Signaling by NOTCH”, R-HSA-194315 “Signaling by Rho GTPases” and R-HSA-195258 “RHO GTPase Effectors” (Appendix A).

## 4. Discussion

We searched for studies concerning HIV-1 reactivation experiments which aimed at reservoir elimination. We obtained RNA-seq reads from 45 samples of human CD4+ T cells across six studies, realigned to the reference human genome, counted, and normalized expression hits, then obtained lists of DEGs in response to LRA treatment. Our goal was to search for genes that could be used for optimizing HIV-1 reactivation, since there has been slow progress in “shock-and-kill” strategies. Re-analyzing studies with different LRAs could provide clues to common genes modulated during HIV-1 reactivation, therefore new strategies could be sought with these candidates in mind. To this end, we performed a cross-reference with other studies from the literature: genome-wide siRNA screens to identify factors involved with HIV-1 replication and life cycle [27,28,29,30], and two databases, HIV-1 Human Protein Interaction Database [17] and RNAcentral [34].

We observed that among the nine LRAs strategies, two (AS1842856 and IL-15) did not significantly modulate gene expression during reanalysis. Three strategies: romidepsin, AZD5582 and SAHA modulated the expression of the most genes across studies. Therefore, we will mostly focus the discussion on them. The other four strategies modulated far less genes when compared to the top three LRAs. This may explain observations that disulfiram was inefficient to reduce the size of HIV-1 reservoir in a single-arm pilot study [53] as well as bryostatin not affecting the transcription of latent HIV-1 in a double-blind phase I pilot clinical-trial [54]. IL-7 use was even associated with transient expansion of HIV-1 reservoir in two independent studies [55,56]. However, it is important to highlight that modulating more genes does not necessarily translate to more efficiency in reservoir reactivation either; indeed it could lead to generalized toxicity or activation of the immune system due to activation of several “off-target” genes [44].

Romidepsin is a histone deacetylase inhibitor that was already investigated in latency reactivation clinical trials. In a small single-arm protocol, romidepsin administration was safe and induced HIV-1 transcription resulting in detectable plasma HIV-1 RNA in patients with long-term viral suppression [57]. Indeed, our results seem to provide a mechanistic evidence for this efficiency, since romidepsin activated the most genes across the studies included in the reanalysis, which would favor HIV-1 reemergence among latently infected T cells. In a following phase 1B/2A protocol, the combination of romidepsin and an experimental HIV-1 vaccine resulted in a 38% reduction in the size of the HIV-1 reservoir from baseline to eight weeks after treatment administration, but it did not seem to prolong median time to virus rebound during ART interruption, meaning that further optimizations in the shock-and-kill strategy are necessary before it can translate to clinical practice [58].

The molecule AZD5582 was tested in macaque and humanized mice alongside ex vivo CD4+ T cells. It robustly reactivated HIV-1 and simian immunodeficiency virus SIV latency from resting CD4+ T cells systemically in those animal models. AZD5582 efficiently inhibits cellular inhibitor of apoptosis protein 1 (cIAP1), a strong repressor of non-canonical NF-κB [44].

SAHA has lower potency than most other HDACi. SAHA in vitro half maximal inhibitory concentration (IC_50_) is on the micromolar range, whereas romidepsin IC_50_ is on the nanomolar range [59]. A previous study compared the HIV-1-reactivation potential of five HDACi (panobinostat, givinostat, belinostat, SAHA and valproic acid) and found that SAHA had potency superior only to valproic acid [60]. This may explain why SAHA modulated less genes, especially in the Mohammadi et al. [40] and Golumbeanu et al. [41] studies, but nonetheless, several were in common with romidepsin.

Beliakova-Bethell et al. [42] stimulated CD4+ T cells with SAHA or romidepsin and compared the modulated genes by these two agents with the HIV-1 Human Protein Interaction Database and obtained a match of 64 genes overall (up- and down-regulated by both SAHA and romidepsin and up- and down-regulated by either of them alone). The genes *CDK2*, *HSPA2, TNF* and *RCOR2* were up-regulated and the genes *IL16, ITK, MX2, NCOA2* and *PIK3CG* were down-regulated by both romidepsin and SAHA and were present in our intersection list. These genes are HIV-1 activators, except for, *IL16, MX2* and *PIK3CG* which are HIV-1 repressors, as curated by Beliakova-Bethell et al. [42].

The concordance between our results and the genome-wide siRNA screens [27,28,29,30] was small, as only 33 unique genes matched our intersection list, within the expected under the binomial test null hypothesis. However, our intersection list had a higher than expected agreement with the HIV-1 Human Interaction Database [31,32,33] following a binomial distribution. Therefore, we have confidence that our intersection list is representative of the true subset of genes involved in HIV-1 reactivation during treatment with LRAs.

One of the reasons “shock-and-kill” treatments do not work as expected may due to post-transcriptional blocks of HIV-1 reactivation, which limits the de novo production of viral particles [40], hampering the detection by the immune system. The role of non-coding RNAs in cellular post-transcriptional regulation is increasingly being investigated. Several classes of ncRNAs were characterized: lcRNAs and miRNAs the best examples, and they may have an influence in the interplay between HIV-1 and host cellular machinery [61].

For example, the *NEAT1* and *NRON* lncRNAs were associated with HIV-1 latency establishment by in vitro studies [62,63]. Host miRNAs, such as *MIR28*, *MIR125B*, *MIR150*, *MIR223*, and *MIR382* repress HIV-1 mRNAs in resting CD4+ T cells [64,65]. Interestingly, *NRON* was modulated by SAHA and romidepsin, perhaps confirming that the cells treated with those LRAs were indeed in a reactivation/proliferation internal milieu, suppressing the regulation pathways brought upon by these RNA molecules.

Other miRNA with evidence pointing to a role in HIV-1 reactivation was detected by our methodology: *MIR155HG*. The function of this miRNA is to suppress the expression of *TRIM32*. HIV-1 trans-activator protein (Tat) sequesters TRIM32 into cytoplasmatic bodies to favor viral expression. Moreover, TRIM32 participates in NF-κB pathway activation. Thus, *MIR155HG/MIR155* expression following reactivation by LRAs may in fact antagonize (“backfire”) its own process of shock-and-kill [65,66]. Importantly, *MIR155HG* was activated precisely by AZD5582, which is associated with NF-κB pathway activation. *TRIM32* itself was not present in our DEGs list, but other 25 related genes were. *TRIM14,* for example, was modulated by romidepsin, SAHA and AZD5582.

Moreover, we observed that SAHA up-regulated the expression of *TCF4*, the gene of a downstream effector of the Wnt/β-catenin pathway that represses both basal and Tat-mediated HIV-1 transcription in astrocytes [67], perhaps by sequestering Tat in the cytoplasm [68,69]. This means that LRAs also “backfire” by up-regulating Wnt pathways in a similar manner to NF-κB pathway activation as discussed above. Notch and Wnt pathways are interconnected in some cellular processes, with Notch being associated with T cell survival and quiescence [70], unfavorable outcomes in the shock-and-kill concept.

The MAPK signaling pathway is exploited by HIV-1 to activate its transcription via translocation of NF-κB to the nucleus and increase in the synthesis of positive transcription elongation factor b (P-TEFb), a key player in HIV-1 genome transcriptional elongation [71]. However, MAPK pathways may protect β-catenin by inhibiting the glycogen synthase kinase 3β (GSK3β) enzyme, which degrades the former in the cytoplasm [69]. We identified enrichment in certain MAPK pathways. Incidentally, MAPK pathways crosstalk with the extracellular-signal-regulated kinase (ERK) pathway. Evidence show that HIV-1 promotes sustained activation of MAPK pathways; this in turn elicits a strong, transient ERK activation, which leads to expression of TNF-α which leads to inflammation and FasL up-regulation, which leads to apoptosis, a desirable outcome for “shock-and-kill” strategy. However, it also may lead to the expression of PD-1, a marker of T cell exhaustion [72], which is undesirable since it suppress T cell cytotoxicity against infected/HIV-1-reactivated cells [73].

*RUNX1* and related genes were modulated by romidepsin (*RUNX1-IT1* and *RUNX2*) and AZD5582 (*RUNX1-IT1*). The RUNX1 protein is a member of a transcription factor family that binds the cofactor CBF-β, to form a transcription factor complex involved in hematopoiesis and CD4+ T cell differentiation. An in vitro study suggests that RUNX1 is involved in HIV-1 latency establishment and a treatment with a RUNX1 inhibitor can reverse the latency [74].

HIV-1 proteins can promote changes in the actin organization, the most abundant cell protein and major constituent of the cytoskeleton. They up-regulate actin regulators gene expression or indirectly activate them. For example, HIV-1 can manipulate Rho-GTPases, a family of proteins that regulate processes of cell morphology change and migration [75]. Interestingly, our reanalysis detected the up-regulation of genes associated with Rac1 and Cdc42 enzymes, major regulators of the cytoskeleton dynamics [75,76] (especially *CDC42EP2*, activated by romidepsin, SAHA and AZD5582 in parallel).

We recognize some limitations our study. First, the data are coming from studies with different sources of cells (from long-term treated patients [40,43,44] or infected in vitro [41,42]). Thus, effects coming from differences in the clonality of the reservoir exposure to antiretroviral drugs, may possibly bias gene expression profiles. Second, the process of HIV-1 latency establishment and its reactivation are highly stochastic phenomena. It is possible that the LRAs only activating a fraction of the cells, which could also introduce bias into RNA expression profiles [77,78].

However, we believe that our cross-referencing helped gauge the relative contribution of each LRA in the patterns of gene modulation and correct (at least some part of) the bias introduced. Second, and we share this limitation with every study of gene expression, is that modulation of transcription does not perfectly correlates with protein abundance, due to the cellular post-transcriptional and translational/post-translational regulatory processes [79]. Therefore, we tried to not become too speculative about the results. Thus, we selected pathways to present in the discussion that already have recognized roles in HIV replication and life cycle, to be as close to a biological significance as possible.

## 5. Conclusions

The methodology of RNA-Seq contributed to increased gene pathway discovery pertaining to major cellular processes, including HIV-1 infection, life cycle and reactivation. Latency reversal is one of the most sought-after method for a functional cure of HIV-1 infection, as ART is incapable of eradicate the virus from the organism of those infected. In our study, we performed a reanalysis of latency reversal strategies and provided a catalog of genes modulated by several LRAs. Although the overlap between the genes modulated by each LRA may seem small, it could help future “shock-and-kill” strategies due to a more profound knowledge of gene pathways involved. Indeed, we could identify common pathways with already observed roles in HIV-1 replication (about 230 genes curated in a specialized database), about 100 non-coding RNA genes among the genes and linked them to the three most prolific LRAs (AZD5582, romidepsin and SAHA), in quantitative terms (number of modulated genes). Therefore, we have confidence that our reanalysis truly reflected biologically significant genes in the context of HIV-1 reactivation.

## Figures and Tables

**Figure 1 microorganisms-08-01505-f001:**
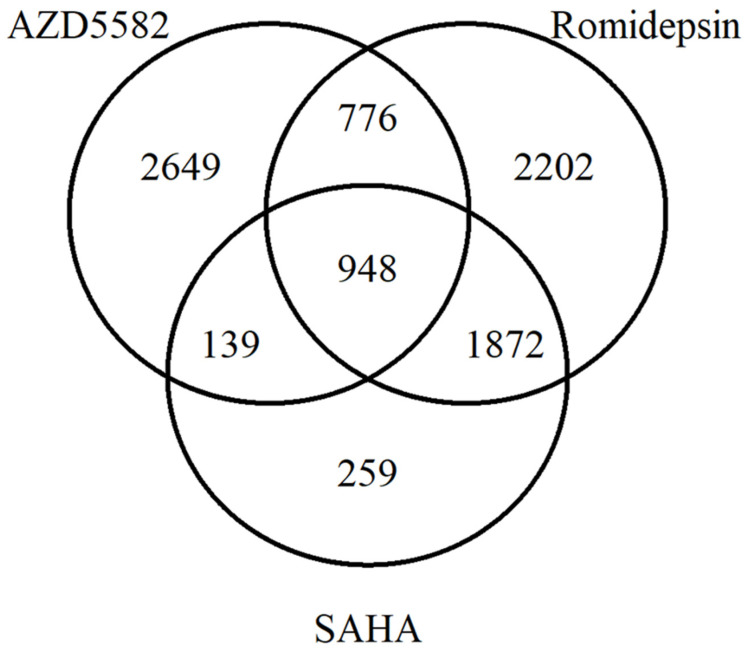
Intersection of differentially expressed genes modulated by romidepsin, SAHA and AZD5582.

**Table 1 microorganisms-08-01505-t001:** Summary of studies included in the reanalysis of gene expression profiles of CD4+ T cells treated with HIV-1 latency reversal agents (LRA).

Study	GSE id	LRA Used	Selected Samples	Control:Treated Ratio	Main Results
Mohammadi et al. (2014) [40]	GSE95297	disulfiram, IL-7, SAHA	4	1:3	Use of SAHA, disulfiram or IL-7 do not result in efficient viral protein expression upon reactivation. The authors suggest that post-transcriptional blocks also contribute to latency.
Golumbeanu et al. (2018) [41]	GSE111727	SAHA	2	1:1	Latently infected cells respond to virus reactivation in a heterogeneous manner. The authors identified a 134-gene cluster in cells most susceptible to induction.
Beliakova-Bethell et al. (2019) [42]	GSE114883	romidepsin or SAHA	12	1:2	The authors identified genes modulated by SAHA and romidepsin that were implicated as HIV transcriptional regulators.
Kulpa et al. (2019) [43]	GSE94150	Bryostatin or IL-15	9	1:2	The authors conclude that HIV gene expression is associated with up-regulation of CD4+ T cell differentiation, acquisition of effector function, and cell cycle entry, being effector memory cells the most inducible cells of the viral reservoir.
Vallejo-Gracia et al. (2019, unpublished)	GSE129522	AS1842856	6	1:1	Not available.
Nixon et al. (2020) [44]	GSE142774	AZD5582 or Ingenol B	12	1:2	The authors suggest that activation of the non-canonical NF-κB signaling pathway by AZD5582 results in the reactivation of HIV in animal models with little toxicity.

**Table 2 microorganisms-08-01505-t002:** Summarized results of the differential expression reanalysis.

Reference	LRA	DEGs ^a^	Up-Regulated Genes	Down-Regulated Genes	Total Expressed Genes	% of DEGs among Expressed Genes	% of DEGs among Whole Genome ^c^
Beliakova-Bethell et al. 2019 [42]	romidepsin	5798	2934	2864	17223	33.7	20.4
Nixon et al. 2020 [44]	AZD5582	4512	1996	2516	15063	30.0	15.9
Beliakova-Bethell et al. 2019 [42]	SAHA ^b^	3218	1651	1567	16949	19.0	11.3
Mohammadi et al. 2014 [40]		942	691	251	17160	5.5	3.3
Golumbeanu et al. 2018 [41]		348	191	157	15991	2.2	1.2
Mohammadi et al. 2014 [40]	disulfiram	473	263	210	17002	2.8	1.7
Nixon et al. 2020 [44]	ingenol B	443	292	151	14935	3.0	1.6
Mohammadi et al. 2014 [40]	IL-7	199	122	77	16503	1.2	0.7
Kulpa et al. 2019 [43]	bryostatin	20	2	18	12511	0.2	0.1
Vallejo-Gracia et al. 2019, unpublished	AS1842856	0	0	0	16840	0.0	0.0
Kulpa et al. 2019 [43]	IL-15	0	0	0	12271	0.0	0.0

^a^ 9494 unique genes, ^b^ 3922 unique genes, ^c^ 28,395 annotated genes.

**Table 3 microorganisms-08-01505-t003:** Distribution of how many genes were activated by number of HIV latency reversal agents (n = 7 strategies).

Number of LRAs	Unique Genes	Percentage
1	5105	53.8
2	3089	32.5
3	1132	11.9
4	152	1.6
5	16	0.2
6 or 7	0	0.0
**Total**	**9494**	**100.0**

**Table 4 microorganisms-08-01505-t004:** Observed and expected intersection of the genes lists of other studies involving HIV replication and life cycle, alongside HIV Human Interaction Database and RNAcentral, a database of non-coding RNA, with the genes modulated by AZD5582, romidepsin and SAHA list (*n* = 948 genes). The binomial distribution was used to test the null hypothesis that the intersection of the differentially expressed genes and other reported genes is equal or less than expected by chance via one-sided tests. The expected intersection was calculated through simulation studies involving the genome size (*n* = 28,395 annotated gene) and the sizes of the two gene lists pairwise compared.

Reference or Database	Unique Genes	Observed Intersection	Expected Intersection	*p*-Value
Konig et al. 2008 [28] ^a^	406	14	13	0.43
Zhou et al. 2008 [29]	264	9	9	0.55
Yeung et al. 2009 [30]	262	11	9	0.29
HIV-1 Human Interaction Database [31,32,33] ^b^	4667	233	156	<0.001
RNAcentral [34] ^b^	7972	110	266	1.00

^a^ includes some genes detected by the Brass et al. 2008 study [27], ^b^ Data retrieved as of 13 August 2020.

**Table 5 microorganisms-08-01505-t005:** Partial results from the gene ontology (GO) analysis of differentially expressed genes modulated by AZD5582, romidepsin and SAHA (*n* = 948 genes) identified through reanalysis of gene expression profiles of CD4+ T cells treated with HIV-1 latency reversal agents (top 20 terms among 278 enriched terms, sorted by false discovery rate p-value from over-representation test).

Pathways and Ranks	GO id	Term	FDR-Adjusted *p*-Value
*Cell activation*			
33	GO:0042110	T cell activation	1.03 × 10^−8^
46	GO:0001775	cell activation	8.97 × 10^−8^
76	GO:0002694	regulation of leukocyte activation	1.44 × 10^−6^
*Cell differentiation*			
16	GO:0000902	cell morphogenesis	2.59 × 10^−10^
38	GO:0030154	cell differentiation	3.19 × 10^−8^
94	GO:0000904	cell morphogenesis involved in differentiation	4.35 × 10^−6^
*Cell proliferation*			
43	GO:0032943	mononuclear cell proliferation	7.36 × 10^−8^
112	GO:0032944	regulation of mononuclear cell proliferation	1.02 × 10^−5^
135	GO:0008283	cell population proliferation	2.54 × 10^−5^
*Cell signaling*			
7	GO:0023052	signaling	4.82 × 10^−14^
12	GO:0007165	signal transduction	9.96 × 10^−12^
15	GO:0023051	regulation of signaling	1.25 × 10^−10^
*MAPK*			
212	GO:0043405	regulation of MAP kinase activity	0.000246
219	GO:0043407	negative regulation of MAP kinase activity	0.000305
*Rho GTPases*			
62	GO:0007266	Rho protein signal transduction	4.87 × 10^−7^
75	GO:0035023	regulation of Rho protein signal transduction	1.35 × 10^−6^

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
