# Peer review of "Reanalysis of Gene Expression Profiles of CD4+ T Cells Treated with HIV-1 Latency Reversal Agents"

_microorganisms, 2020, doi:10.3390/microorganisms8101505_

Round 1

Reviewer 1 Report

Our understanding of the approach adopted in this study is that it lumped together RNA-seq datasets from untreated samples across 6 different studies, and used it as control for the RNA-seq datasets of LRA-treated samples from the same 6 studies. If this is correct, the analysis—while statistically strong—it is not biologically sound. That is because it utilizes datasets generated under very different conditions. It compares cells from HIV-1 infected patients and cells from healthy donors that are infected in vitro. It compares latently-infected cells generated after in vitro infection under very different conditions (for method of cell activation, for virus used for the infection, for method of expansion, and for method to achieve cell quiescence and viral latency). In this situation, the LRA-treated samples derived from a single study (for instance the samples treated with IL-15 from the Kulpa et al. study) are compared to the average of the negative (untreated) controls from multiple and very different studies, possibly revealing differences that with a more uniform and stringent control group (such as the one in the Kulpa et al. study itself) would not emerge.

The fact that in the meta-analysis 318 DEGs (20% of the total) were modulated by LRAs that could not be identified is very worrisome. The Authors argue that this is due to an increase in the power of the meta-analysis. An alternative explanation is that this meta-analysis lumps together very different samples and reveals incorrect differences in gene expression.

A more rational approach would be to extract DEGs from each study, and compare them with DEGs from other studies to identify common DEGs across all studies. The analysis would be performed across datasets rather than by lumping together datasets.

One key aspect of all clinically relevant LRAs is that they reverse HIV-1 latency without inducing T cell activation. PMA and ionomycin are not clinically relevant latency reversing agents as they are toxic and cause global T cell activation. At the same time, CD3+CD28+IL-2 stimulation is also not clinically relevant because of its toxicity and effects on global T cell activation. These two stimulation methods (PMA+ionomycin and CD3+CD28+IL-2) were used as “positive controls”. In other words, they were used as a way of comparing the effects of LRAs to stimuli that achieve maximal level of HIV-1 activation. Unstimulated cells (or vehicle-treated cells, such as DMSO-treated cells) are “negative controls”.

The term “ex vivo” (line 78) is not used accurately. By ex vivo it is meant cells from HIV-1 infected individuals, namely cells that were infected in an actual patient (as in the study by Mohammadi et al.; line 183). On the contrary, cells obtained from HIV-1 negative donors that are infected in the laboratory (as in the study by Beliakova-Bethell et al.; line 196) represent an in vitro model even though they are primary cells. There is a significant difference in behavior between primary CD4+ T cells obtained from HIV-1 patients and CD4+ T cells obtained from uninfected donors, which are then infected in vitro. Therefore, the analysis in the present study compares “apples to oranges”.

Lines 434-445 provide the rationale as to why the meta-analysis was representative of the individual analyses. While this explanation is “statistically” sound, it is not “biologically” sound because the studies used for the meta-analysis were conducted under very different conditions.

Ultimately, this meta-analysis fails in two ways: it does not identify a pathway/gene that is activated in response to stimulation with all LRAs; and it does not identify previously unknown pathways/genes activated in response to stimulation with one or more LRAs. This meta-analysis is merely confirmatory of previously known facts.    

The English in the manuscript would benefit by revision from a native English speaker. Some words appear to be a “literal” translation, but the meaning of the translated word is inaccurate. An example is the word “foresee” (line 13), which means “predict” or “forecast”. Presumably the Authors used a literal translation of a word in their native language, which however does not fit the meaning in that sentence. Another example is the word “analysis” (line 94), which should be plural (analyses).

Author Response

Our understanding of the approach adopted in this study is that it lumped together RNA-seq datasets from untreated samples across 6 different studies, and used it as control for the RNA-seq datasets of LRA-treated samples from the same 6 studies. If this is correct, the analysis—while statistically strong—it is not biologically sound. That is because it utilizes datasets generated under very different conditions. It compares cells from HIV-1 infected patients and cells from healthy donors that are infected in vitro. It compares latently-infected cells generated after in vitro infection under very different conditions (for method of cell activation, for virus used for the infection, for method of expansion, and for method to achieve cell quiescence and viral latency). In this situation, the LRA-treated samples derived from a single study (for instance the samples treated with IL-15 from the Kulpa et al. study) are compared to the average of the negative (untreated) controls from multiple and very different studies, possibly revealing differences that with a more uniform and stringent control group (such as the one in the Kulpa et al. study itself) would not emerge.

The fact that in the meta-analysis 318 DEGs (20% of the total) were modulated by LRAs that could not be identified is very worrisome. The Authors argue that this is due to an increase in the power of the meta-analysis. An alternative explanation is that this meta-analysis lumps together very different samples and reveals incorrect differences in gene expression.

A more rational approach would be to extract DEGs from each study, and compare them with DEGs from other studies to identify common DEGs across all studies. The analysis would be performed across datasets rather than by lumping together datasets.

Response (to all points above): Thank you for your suggestions. We extracted DEGs from each study and compared them with DEGs from other studies to identify common DEGs across all studies as suggested. As such, our methodology changed to a comparison of DEGs lists between studies, as the meta-analysis was not adequate as pointed by the reviewer. Thus, we performed extensive modifications in the Material and Methods, Results, and Discussion sections of the manuscript, as well as changing the title to better reflect the new approach of data analysis.

One key aspect of all clinically relevant LRAs is that they reverse HIV-1 latency without inducing T cell activation. PMA and ionomycin are not clinically relevant latency reversing agents as they are toxic and cause global T cell activation. At the same time, CD3+CD28+IL-2 stimulation is also not clinically relevant because of its toxicity and effects on global T cell activation. These two stimulation methods (PMA+ionomycin and CD3+CD28+IL-2) were used as “positive controls”. In other words, they were used as a way of comparing the effects of LRAs to stimuli that achieve maximal level of HIV-1 activation. Unstimulated cells (or vehicle-treated cells, such as DMSO-treated cells) are “negative controls”.

Resposnse: we removed PMA+ionomycin and CD3+CD28+IL-2 stimulation from analysis, maintaining just the other interventions.

The term “ex vivo” (line 78) is not used accurately. By ex vivo it is meant cells from HIV-1 infected individuals, namely cells that were infected in an actual patient (as in the study by Mohammadi et al.; line 183). On the contrary, cells obtained from HIV-1 negative donors that are infected in the laboratory (as in the study by Beliakova-Bethell et al.; line 196) represent an in vitro model even though they are primary cells. There is a significant difference in behavior between primary CD4+ T cells obtained from HIV-1 patients and CD4+ T cells obtained from uninfected donors, which are then infected in vitro. Therefore, the analysis in the present study compares “apples to oranges”.

Response: see response below.

Lines 434-445 provide the rationale as to why the meta-analysis was representative of the individual analyses. While this explanation is “statistically” sound, it is not “biologically” sound because the studies used for the meta-analysis were conducted under very different conditions.

Response: we included a paragraph into the Discussion section to address the limitations pointed in the previous items: “We recognize some limitations our study. First, the data are coming from studies with different sources of cells (from long-term treated patients [40,43,44] or infected in vitro [41,42]). Thus, effects coming from differences in the clonality of the reservoir exposure to antiretroviral drugs, may possibly bias gene expression profiles. Second, the process of HIV-1 latency establishment and its reactivation are highly stochastic phenomena. It is possible that the LRAs only activating a fraction of the cells, which could also introduce bias into RNA expression profiles [77,78].”

Ultimately, this meta-analysis fails in two ways: it does not identify a pathway/gene that is activated in response to stimulation with all LRAs; and it does not identify previously unknown pathways/genes activated in response to stimulation with one or more LRAs. This meta-analysis is merely confirmatory of previously known facts.

Response: with our new approach, we believe that genes and pathways are better associated with each LRA. Even if our work is merely confirmatory of previously known facts it has importance because it offers a centralization of current knowledge, helping future investigators.

The English in the manuscript would benefit by revision from a native English speaker. Some words appear to be a “literal” translation, but the meaning of the translated word is inaccurate. An example is the word “foresee” (line 13), which means “predict” or “forecast”. Presumably the Authors used a literal translation of a word in their native language, which however does not fit the meaning in that sentence. Another example is the word “analysis” (line 94), which should be plural (analyses).

Response: we performed the suggested alterations.

Reviewer 2 Report

Meta-analysis of gene expression profiles of CD4+ T 2 cells treated with HIV-1 latency reversal agents by Antonio Victor Campos Coelho et al.

The authors sought to find possible common molecular mechanisms of reactivation of latent HIV-1 in the viral reservoir by latency reversal agents (LRAs). They hypothesized that this would advance the field by finding new, more efficient, “shock-and- kill” strategies. For this purpose they re-analyzed RNA-Seq assays to find differentially expressed genes (DEGs) during latency reversal via transcriptome analysis. LRA treatments resulted in enrichment of seven pathways relevant to HIV-1 life cycle.

This is an interesting manuscript advancing our knowledge related to HIV-1 latency reversal. This is a very important field of study. The paper is well-written, easy to read even for someone outside of this research area. The methodology is appropriate and the results are convincing. The weaknesses of the methods are also discussed in detail. It is a balanced paper clearly presenting the pros and cons of the methods and results.

Author Response

The authors sought to find possible common molecular mechanisms of reactivation of latent HIV-1 in the viral reservoir by latency reversal agents (LRAs). They hypothesized that this would advance the field by finding new, more efficient, “shock-and- kill” strategies. For this purpose they re-analyzed RNA-Seq assays to find differentially expressed genes (DEGs) during latency reversal via transcriptome analysis. LRA treatments resulted in enrichment of seven pathways relevant to HIV-1 life cycle.

This is an interesting manuscript advancing our knowledge related to HIV-1 latency reversal. This is a very important field of study. The paper is well-written, easy to read even for someone outside of this research area. The methodology is appropriate and the results are convincing. The weaknesses of the methods are also discussed in detail. It is a balanced paper clearly presenting the pros and cons of the methods and results.

Response: thank you for your review.

Reviewer 3 Report

The paper by Coelho et al examines data from six previous studies which HIV-1 infected primary cells are treated with different latency reversal agents. Their examination of the data sets demonstrates changes in the expression for several genes and suggest a critical role for numerous pathways including the Rho, MAPK, Wnt, Notch and NF-kB pathways. The scientific premise for the study to re-examine data from previous studies to gain insights into commonalities of different compounds and different reversal strategies is strong and there is a need for these studies that attempt to integrate pre-existing large data sets. The authors do make the effort to discuss in detail the different gene sets and provide a holistic view of latency reversal in both the similarities and heterogeneity although some of the discussion gets weighed down with peripheral details of negative data, individual genes and tangential events, for example the discussion of Vpr and DNA repair. The paper also should more critically evaluate the different models and LRAs. The data are coming from long-term treated patients and in vitro infections and the number of infected cells, the clonality of the reservoir and exposure to ART, would be expected to be very different and may possibly bias gene expression profiles. Furthermore, there should be more detailed discussion about the efficacy of the latency reversal agents and how might this be biasing the different data sets. For example, the data supporting the ability of Disulfiram to reactivate latent infection is not particularly strong in vitro and in vivo it has been shown not to be effective (Spivak et al). Even the best LRAs are probably only activating a fraction of the cells suggesting a stochastic aspect of latency and T cell activation. This needs to be addressed.

Author Response

The paper by Coelho et al examines data from six previous studies which HIV-1 infected primary cells are treated with different latency reversal agents. Their examination of the data sets demonstrates changes in the expression for several genes and suggest a critical role for numerous pathways including the Rho, MAPK, Wnt, Notch and NF-kB pathways. The scientific premise for the study to re-examine data from previous studies to gain insights into commonalities of different compounds and different reversal strategies is strong and there is a need for these studies that attempt to integrate pre-existing large data sets. The authors do make the effort to discuss in detail the different gene sets and provide a holistic view of latency reversal in both the similarities and heterogeneity although some of the discussion gets weighed down with peripheral details of negative data, individual genes and tangential events, for example the discussion of Vpr and DNA repair. The paper also should more critically evaluate the different models and LRAs. The data are coming from long-term treated patients and in vitro infections and the number of infected cells, the clonality of the reservoir and exposure to ART, would be expected to be very different and may possibly bias gene expression profiles. Furthermore, there should be more detailed discussion about the efficacy of the latency reversal agents and how might this be biasing the different data sets. For example, the data supporting the ability of Disulfiram to reactivate latent infection is not particularly strong in vitro and in vivo it has been shown not to be effective (Spivak et al). Even the best LRAs are probably only activating a fraction of the cells suggesting a stochastic aspect of latency and T cell activation. This needs to be addressed.

Response: we performed extensive modifications in the Material and Methods, Results, and Discussion sections of the manuscript, as well as changing the title to better reflect the new approach of data analysis. We believe the suggested alterations were met.

Round 2

Reviewer 1 Report

The authors have performed an extensive revision of the initial submission. The authors have adequately addressed all the issues raised during the first round of review. In its revised form, the manuscript is significantly improved.